# Alternative Genetic Diagnoses in Axenfeld–Rieger Syndrome Spectrum

**DOI:** 10.3390/genes14101948

**Published:** 2023-10-17

**Authors:** Linda M. Reis, David J. Amor, Raad A. Haddad, Catherine B. Nowak, Kim M. Keppler-Noreuil, Smith Ann Chisholm, Elena V. Semina

**Affiliations:** 1Department of Ophthalmology and Visual Sciences, Medical College of Wisconsin, Milwaukee, WI 53226, USA; lreis@mcw.edu (L.M.R.); sachisholm@mcw.edu (S.A.C.); 2Department of Pediatrics and Children’s Research Institute, Medical College of Wisconsin and Children’s Wisconsin, Milwaukee, WI 53226, USA; 3Murdoch Children’s Research Institute, Department of Paediatrics, University of Melbourne, Parkville, VIC 3052, Australia; david.amor@mcri.edu.au; 4Division of Endocrinology, Diabetes, and Metabolic Diseases, Sidney Kimmel Medical College at Thomas Jefferson University, Philadelphia, PA 19107, USA; 5Division of Genetics and Metabolism, MassGeneral Hospital for Children, Boston, MA 02114, USA; cnowak@mgb.org; 6Department of Pediatrics, University of Wisconsin School of Medicine and Public Health, Madison, WI 53726, USA; kepplernoreu@wisc.edu

**Keywords:** Axenfeld–Rieger anomaly, Axenfeld–Rieger syndrome, *USP9X*, *JAG1*, *CDK13*, *BCOR*, *HCCS*, *AMELX*

## Abstract

Axenfeld–Rieger anomaly (ARA) is a specific ocular disorder that is frequently associated with other systemic abnormalities. *PITX2* and *FOXC1* variants explain the majority of individuals with Axenfeld–Rieger syndrome (ARS) but leave ~30% unsolved. Here, we present pathogenic/likely pathogenic variants in nine families with ARA/ARS or similar phenotypes affecting five different genes/regions. *USP9X* and *JAG1* explained three families each. *USP9X* was recently linked with syndromic cognitive impairment that includes hearing loss, dental defects, ventriculomegaly, Dandy–Walker malformation, skeletal anomalies (hip dysplasia), and other features showing a significant overlap with *FOXC1*-ARS. Anterior segment anomalies are not currently associated with *USP9X*, yet our cases demonstrate ARA, congenital glaucoma, corneal neovascularization, and cataracts. The identification of *JAG1* variants, linked with Alagille syndrome, in three separate families with a clinical diagnosis of ARA/ARS highlights the overlapping features and high variability of these two phenotypes. Finally, intragenic variants in *CDK13*, *BCOR*, and an X chromosome deletion encompassing *HCCS* and *AMELX* (linked with ocular and dental anomalies, correspondingly) were identified in three additional cases with ARS. Accurate diagnosis has important implications for clinical management. We suggest that broad testing such as exome sequencing be applied as a second-tier test for individuals with ARS with normal results for *PITX2/FOXC1* sequencing and copy number analysis, with attention to the described genes/regions.

## 1. Introduction

Axenfeld–Rieger anomaly/syndrome (ARA/ARS) is a specific type of anterior segment disorder characterized by the triad of posterior embryotoxon (prominent annular white line near the limbus at the level of the Descemet membrane), iridocorneal adhesions, and iris anomalies including iris hypoplasia, corectopia, and/or polycoria [1,2]. The ocular features are frequently associated with additional nonocular anomalies with two distinct but highly variable syndromes described. In *PITX2*-assocated ARS (type 1), the ocular anomalies are almost universally associated with dental (missing and malformed teeth) and umbilical (redundant umbilical skin, umbilical hernia, and/or omphalocele) features [3]. *FOXC1*-associated ARS (type 3) causes much more variable nonocular features, with some individuals having isolated ocular features and others affected with a variety of additional anomalies including hearing loss, congenital heart defects, enamel hypoplasia/dental crowding, hip dysplasia and other skeletal anomalies, hypotonia/early delay, and white matter hyperintensities [3]. At the most severe end of the spectrum for *FOXC1* disruption is De Hauwere syndrome, characterized by anterior segment anomalies, hypertelorism, short stature, hearing loss, hydrocephalus, joint hyperlaxity, and skeletal anomalies including hip dysplasia [3,4,5]. While Axenfeld–Rieger anomaly (ARA) is the most common ocular diagnosis in both types of ARS, a wide range of anterior segment ocular phenotypes have been reported for both genes but especially for *FOXC1* [3]. Approximately 70% of ARS can be explained by sequence or copy number variants affecting *PITX2* or *FOXC1*, indicating that additional genetic causes remain to be elucidated [3].

Other genetic variants have been occasionally reported in individuals with a clinical diagnosis of ARS or isolated ARA, including dominant variants in *COL4A1* in multiple families [6,7], recessive variants in *CPAMD8*, *ADAMTS17*, and *CYP1B1* reported in one family each [8,9], and dominant variants in histone methyltransferases *SETD1A* and *KMT2F* in one family each [10]. Large deletions affecting 13q were reported in three individuals with ARA with no causative gene identified [11,12,13]. *PAX6*, typically associated with aniridia, can present with overlapping anterior segment ocular phenotypes but is not a cause of ARA specifically [14].

## 2. Materials and Methods

This human study was approved by the Institutional Review Boards of the Medical College of Wisconsin and University of Iowa. We reviewed our cohort of unexplained individuals with ARA/ARS or similar phenotypes without sequence or copy number variants in *PITX2* or *FOXC1* [3]. Exome sequencing was performed by Psomagen (Rockville, MD, USA) and analyzed using VarSeq™ (Golden Helix, Bozeman, MT, USA), as previously described [10]. We reviewed variants with a frequency <0.001 in gnomAD v2.1.1 for pathogenic/likely pathogenic variants in all other genes associated with anterior segment disorders and other OMIM genes associated with pediatric syndromes. Copy number analysis was performed on exome data using the VS-CNV^®^ caller. Pathogenicity was determined using ACMG/AMP criteria [15,16]. Variants were confirmed by Sanger sequencing (research and/or clinical confirmation), TaqMan assays (clinical), or independent exome analysis. Parents were tested when available by either exome analysis or Sanger sequencing.

## 3. Results

### 3.1. Identification of UPS9X and JAG1 Variants in Multiple Families with ARS

Pathogenic/likely pathogenic variants were identified in nine families affecting five genes (Table 1). All variants were absent from gnomAD. Two genes, *UPS9X* and *JAG1*, explained multiple cases within our cohort. The most novel of the two was *USP9X*, with three unrelated individuals carrying causative variants in this gene, which has not been recognized as a cause of anterior segment phenotypes.

Individual 1 is a 14-year-old White (U.S.) female diagnosed with ARS by her ophthalmologist. Ocular anomalies include ARA, early signs of glaucoma, and myopia. Nonocular anomalies include choanal atresia with deviated nasal septum, high arched palate with bilateral membranous palate anomaly, chronic sinusitis and otitis media (six sets of ear tubes, ear drum reconstruction, and subsequent mild hearing loss), dysmorphic facial features (long, thin face; down-slanting, short palpebral fissures; long, prominent nose that is thin at the nasal tip and pinched at the alar region), abnormal skin pigmentation along lines of Blaschko, and precocious puberty (menses at 9 years old). She has a history of early delays, especially gross motor, but transitioned out of intervention by first grade and is now on the honor roll with adjusted workload, and a history of depression. Height, weight, and head circumference are all normal (50th centile). Exome sequencing identified a likely pathogenic variant in *USP9X*, NM_001039590.2:c.1314+2T>C. Parental samples were not available. In silico analysis revealed a strong prediction of loss of the donor splice site (0.73 by SpliceAI). There is a weak prediction (0.04) of donor gain which would result in 91bp deletion from exon 10. Exon skipping would result in 153bp deletion of exon 10, c.1162_1315del p.(Glu388_Gln438del). Exon 10 is highly conserved with a very low rate of benign missense variants in gnomAD, so this change is likely to be deleterious.

Individual 2 (Figure 1A) is a 4-year-old Black (U.S.) female with features overlapping ARS including anterior segment dysgenesis, maxillary hypoplasia, unilateral sensorineural hearing loss, congenital heart defect (bicuspid and dysplastic aortic valve), enamel hypoplasia with severe caries and two congenitally missing secondary teeth, umbilical hernia, left developmental hip dysplasia, and Dandy–Walker malformation. Specific ocular anomalies in the left eye include buphthalmos; opaque, thickened, vascularized cornea; and corneal ectasia with spontaneous corneal perforation at 6 months followed by enucleation. Pathology revealed thickened cornea with hypercellular fibrosis and vascularization, cataractous changes in the lens, and only a small remnant of the ciliary body and iris, along with retinal atrophy with thinning of the inner nuclear layer. The right eye has a history of corneal ulcer with residual scarring, hypopigmented wedge in iris, and mild myopia (−1.75). Additional nonocular anomalies include small, posteriorly rotated ears; broad nose with short columella and pointed, depressed tip; flat philtrum; brachycephaly; low anterior hairline; and bilateral coxa valga. She has a history of poor growth with height < 1st centile and weight 3–5th centile and significant global developmental delay (nonverbal). Exome sequencing identified a likely pathogenic variant in *USP9X*, c.121G>T; p.(Glu41*), which was not present in the mother (father not available).

Individual 3 is a 9-year-old White (U.S.) female with features overlapping ARS including enamel hypoplasia and dental crowding, bilateral hip developmental dysplasia, congenital heart defects (interrupted aortic arch type B with large ventricular septal defect, left superior vena cava to coronary sinus, and bicuspid aortic valve), and history of feeding difficulties in infancy. Ocular anomalies include phlyctenular keratoconjunctivitis and superficial punctate keratitis, subtle scalloped scarring of the cornea with neovascularization, juvenile cataract (small dot multilayer but mostly cortical, not visually significant), mild myopia (−1.25; −1.00), amblyopia, and blepharokeratitis. Additional nonocular anomalies include high arched palate, low-set posteriorly rotated ears with overfolded helices, frontal bossing, anteverted nares, scoliosis with tethered cord, vesicoureteral reflux, left vocal cord paralysis, hypopigmentation of skin with cutis marmorata, eczema, and gross/fine motor delays but normal cognition. Height and weight are within the normal range (22nd and 70th centile, respectively). Exome sequencing identified a *de novo* likely pathogenic variant in *USP9X*, c.1603dupA p.(Ile535Asnfs*11); she was reported in a previous cohort at 2 years of age [17].

Individual 4 is a 20-year-old White (Australia) female referred with diagnosis of ARS based on ARA, bilateral sensorineural hearing loss, midface hypoplasia, hypertelorism, and dental crowding. Specific ocular anomalies consisted of posterior embryotoxon, iridocorneal adhesions, and iris hypoplasia along with hyperopia. Additional nonocular features included prominent forehead, short philtrum, arachnodactyly, marfanoid habitus, postural orthostatic tachycardia syndrome (POTS), and auditory/visual processing deficit with average cognition. Echocardiogram was normal. Exome sequencing identified a likely pathogenic variant in *JAG1*, NM_000214.3:c.59dupT p.(Leu21Alafs*52). Parental samples were not available.

Individual 5 (Figure 1B) is a 39-year-old White (U.S.) female referred with a diagnosis of ARS including ARA, narrow teeth and hypodontia (missing at least four teeth) along with frequent caries, and ventricular septal defect. Specific ocular anomalies consisted of bilateral Axenfeld anomaly with severe myopia and glaucoma diagnosed at 39 years of age. Additional nonocular anomalies include irritable bowel syndrome, pituitary macroadenoma (prolactinoma), and mildly elevated total bilirubin, AST and ALT. CNV analysis of exome sequencing data identified a pathogenic 3.01 Mb deletion of 20:10256130-13269324 (hg19) including seven genes (*BTBD3*, *ISM1*, *JAG1*, *MKKS*, *SLX4IP*, *SNAP25*, and *SPTLC3*), of which *JAG1* is pathogenic. Parental samples were not available.

Individual 6 (Figure 1C,D) is a 38-year-old female clinically diagnosed with ARS in childhood by her ophthalmologist. She has posterior embryotoxon, iris hypoplasia, significant corectopia, small optic cup, and hyperopia (+4.25, +3.75). Nonocular anomalies include dental enamel defects, heart murmur, and polycystic ovary syndrome. Her 5-month-old son was born with congenital heart defects (pulmonary stenosis, atrial septal defect, and patent ductus arteriosus), neonatal jaundice and biliary atresia, and slightly small left kidney (no eye exam but no obvious iris anomalies). Exome sequencing identified a pathogenic variant in *JAG1*, c.2419G>T p.(Glu807*) in the proband and her son.

### 3.2. Identification of Variants in BCOR and CDK13 and chr X Deletion in Three Unrelated Cases

Variants in two other genes (*CDK13*, *BCOR*) were identified in a single family each, one of which (*CDK13*) had not been previously linked with anterior segment anomalies. Individual 7 is an 11-year-old White (U.S.) female with a clinical diagnosis of ARS including ARA and widely spaced eyes. Specific ocular features include bilateral posterior embryotoxon and mild iris hypoplasia with transillumination defects, megalocornea with normal intraocular pressure, and mild myopia (−1.00 OU). Additional nonocular anomalies include broad nasal tip, small mouth, history of recurrent otitis media, nasal speech, developmental delay, moderate hypotonia, and cognitive impairment. Height and weight were normal (75th–90th centile). Her 14-year-old brother, affected with cognitive impairment, mild hypotonia, hypospadias, cryptorchidism, history of feeding difficulties, nasal speech, bipolar disorder, behavior problems, and mild dysmorphic features (thin upper lip, telecanthus, right short fifth finger) was thought to have a different genetic syndrome from the proband. Both parents have a history of developmental delay with special education services in school but were unavailable for testing. The mother also has schizophrenia and an extensive family history of learning disabilities and psychiatric illness. Exome sequencing identified a likely pathogenic variant in the proband and her brother within the protein kinase domain of *CDK13*, NM_003718.5:c.2252G>A p.(Arg751Gln) with high CADD (31) and REVEL (0.697) scores. The same variant was reported previously in an unrelated individual within the Deciphering Developmental Disorders cohort [18].

Individual 8 is a 20-year-old White (U.S.) female referred with a clinical diagnosis of ARS based on ocular anomalies, peg-shaped and small teeth with dental crowding, maxillary hypoplasia, hearing loss, and redundant periumbilical skin. Specific ocular anomalies include bilateral congenital cataract, glaucoma diagnosed at 3 years of age, optic nerve hypoplasia, mild microphthalmia (left smaller than right), and amblyopia of the left eye. Additional nonocular anomalies include patent foramen ovale, nasal deviation, asymmetric and posteriorly rotated ears, narrow palate with thickened ridges, and fifth finger clinodactyly. Height and weight are normal (10–25th centile), as is cognition. Exome sequencing identified a likely pathogenic variant in *BCOR* NM_001123385.2: c.3350_3360dupCAGACCAGGTG p.(Ala1121GlnfsTer42). Parental samples were not available.

The final case was explained by a combination of two disrupted genes. Individual 9 is a 57-year-old White (U.S.) female who was diagnosed with ARS in early childhood based on ocular features and dental anomalies including small, mispositioned teeth, enamel hypoplasia, and extra teeth. Specific ocular features include bilateral congenital glaucoma with surgery in infancy, polycoria, microphthalmia (right smaller than left), corneal opacity with failed corneal transplant in adulthood, and small congenital cataract which did not require removal until 30 years of age. Additional nonocular anomalies include hearing loss (40s), rheumatoid arthritis, irritable bowel syndrome, connective tissue disease, and migraines. Height/weight and cognitive development are normal. Exome sequencing identified a pathogenic 4.08 Mb deletion of X:7370404-11445756 (hg19) including 17 genes (*AMELX*, *ANOS1*, *ARHGAP6*, *CLCN4*, *CLDN34*, *FAM9A*, *FAM9B*, *GPR143*, *HCCS*, *MID1*, *PNPLA4*, *SHROOM2*, *TBL1X*, *VCX*, *VCX2*, *VCX3B*, *WWC3*), of which *HCCS* and *AMELX* are pathogenic. Parental samples were not available.

## 4. Discussion

Axenfeld–Rieger syndrome is associated with significant variability in both ocular and nonocular features [3]. For *FOXC1* in particular, a wide range of clinical presentations has been reported. While the majority of cases are caused by disruption of the two major genes, *PITX2* and *FOXC1*, a significant portion remains unexplained.

Dominant loss-of-function and missense variants in *USP9X* are well-recognized as a cause of syndromic cognitive impairment [19,20]. Both male and female cohorts have been reported, with females showing a broader spectrum of syndromic anomalies [19]. While anterior segment ocular anomalies are not frequently reported in previous cohorts, the syndromic features of hearing loss, ventriculomegaly, Dandy–Walker malformation, short stature, skeletal anomalies including hip dysplasia, and dental defects show significant overlap with *FOXC1*-associated phenotypes, especially De Hauwere syndrome. Ocular phenotypes including cataract and myopia have been reported in some females, and corneal opacity was noted in one other girl [19,20]. Features unique to *USP9X* include abnormal pigment distribution along lines of Blaschko, choanal atresia, polydactyly, agenesis of the corpus callosum, and a higher rate of cognitive impairment. Interestingly, two of the three females reported here had a history of early delay but with normal cognitive abilities at older ages. Two of the three had significant anterior segment phenotypes consistent with *FOXC1* disruption, and the third has subtle signs of corneal neovascularization, which has also been reported in human patients and mice with *Foxc1* disruption [21]. In support of a role for *USP9X* in the developing eye, *Usp9x* is highly expressed in the mouse eye, similar to the level seen in the brain [22]. *USP9X* has been shown to play a role in the WNT pathway [23,24], which is known to be downstream of *PITX2* and important in the ocular phenotype associated with this gene [25]. Additionally, loss of *Usp9x* resulted in reduced expression of N-cadherin (*Cdh2*) [24], also associated with a corneal phenotype in humans [26].

Alagille syndrome, caused by variants in *JAG1* or *NOTCH2*, is a well-recognized phenotype characterized by highly penetrant posterior embryotoxon and liver dysfunction along with cardiac and vertebral anomalies [27]. While isolated posterior embryotoxon is the characteristic ocular anomaly associated with Alagille syndrome, iris hypoplasia was also present in 20–40% of those undergoing thorough eye exam [28,29]. The diagnosis of Alagille syndrome in three separate families with a clinical diagnosis of ARS was somewhat surprising and highlights the extreme variability of this phenotype. None of the patients meet the characteristic clinical criteria requiring involvement of three out of five organ systems (liver, cardiac, skeletal, ocular, facial). The lack of severe liver involvement likely complicated the clinical recognition of Alagille syndrome in these probands, since bile duct paucity is considered one of the defining features; two of the three individuals reported here had no known liver dysfunction and the third had only mildly abnormal liver function tests discovered through blood work. All three individuals also reported additional features more typical of ARS including dental anomalies (3) and hearing loss (2), though hearing loss has been occasionally reported with *JAG1* variants [30].

Similar to *UPS9X*, disruption of *CDK13* is associated with syndromic cognitive impairment [18,31]. The majority of variants are missense and almost all are clustered within the protein kinase domain of CDK13. Cognitive impairment and behavioral problems are the most consistent features with variable dysmorphic facial features, congenital heart defects, and structural brain anomalies. Eye anomalies are recognized but primarily consist of strabismus, so the association with ARA is novel. Similar to *Usp9x*, expression of *Cdk13* in the mouse eye is seen at a similar level to the brain [22].

*BCOR* is well recognized with loss-of-function variants linked to X-linked dominant Oculofaciocardiodental (OFCD) syndrome in females, while missense variants are occasionally reported in males, typically with a more severe syndromic microphthalmia phenotype [32,33]. OFCD is highly variable and shows substantial overlap with combined features of the two types of ARS including missing teeth (*PITX2*) and hearing loss and congenital heart defects (*FOXC1*); while noted to be rare, umbilical hernia has occasionally been reported as well [34,35]. Congenital cataracts with microphthalmia are highly penetrant in OFCD but less commonly seen in ARS; anterior segment phenotypes including iris synechiae, iris coloboma, and posterior embryotoxon are occasionally reported with *BCOR* variants [33]. Nonocular features specific to *BCOR* include 2–3 toe syndactyly, extra teeth, or radiculomegaly, and developmental delay/cognitive impairment is more common with *BCOR* but highly variable [33].

The final case is an interesting example of a contiguous gene syndrome whose combined features resulted in a phenotype overlapping ARS. *HCCS* was first recognized as a cause of X-linked dominant linear skin defects with multiple congenital anomalies, with microphthalmia, sclerocornea, heart defects, structural brain anomalies, and linear skin defects on the face and neck [36]. Subsequent studies revealed a highly variable phenotype in affected individuals, with isolated ocular features in some females [37]. The addition of *AMELX* within the deleted region resulted in amelogenesis imperfecta, type 1E [38], with significant overlap with the dental features of ARS including severe enamel hypoplasia.

All of the genetic diagnoses identified here showed substantial overlap with the ARS phenotypic spectrum. Accurate diagnosis has important implications for clinical management and family planning. We suggest that broad testing such as exome sequencing be applied as a second-tier test for individuals with ARS with normal sequencing/copy number results for *PITX2/FOXC1*.

## Figures and Tables

**Figure 1 genes-14-01948-f001:**
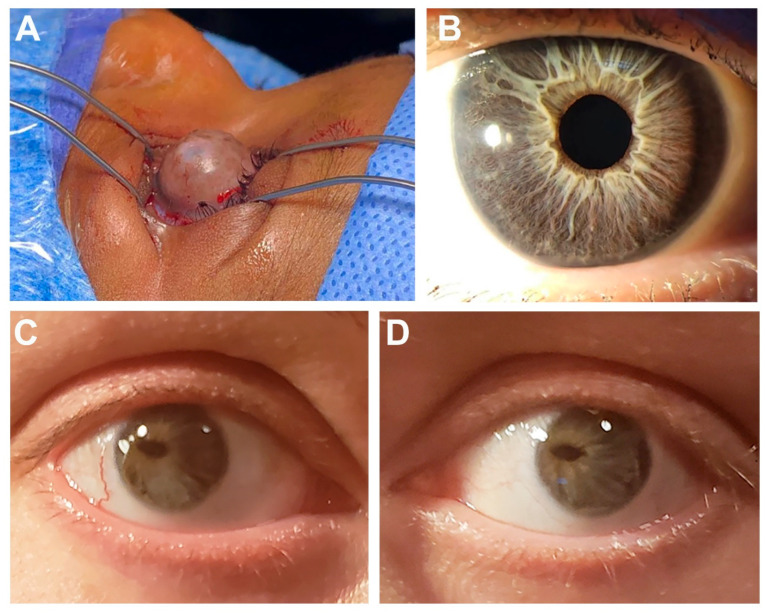
Selected ocular images demonstrating anterior segment phenotypes. (**A**) Right eye of Individual 2 showing corneal ectasia with large bulbous anterior extension. (**B**) Eye of Individual 5 showing posterior embryotoxon and mild iris hypoplasia (**C**,**D**) Bilateral eye images of Individual 6 showing severe corectopia, posterior embryotoxon and iris hypoplasia.

**Table 1 genes-14-01948-t001:** Clinical features of presented individuals.

Individual	Ocular	Other	Gene	Variant	ACMG Criteria
1	**ARA**, **GL**, MY	**Mild delay but normal cognition**, **dysmorphic facial features**, palate anomaly, choanal atresia, lines of Blaschko, precocious puberty	*USP9X*	NM_001039590.2: c.1314+2T>C	LP: PVS1, PM2_supp
2	**Left: ASD**, **severe IH**, **CAT**, **buphthalmos**, **corneal neovascularization**, corneal ectasia with perforation Right: corneal ulcer/scarring, MY, iris hypopigmentation	**DA**, **HL**, **dysmorphic facial features**, **CHD**, **Dandy–Walker malformation**, **hip dysplasia**, **umbilical hernia**, global delay, short stature	*USP9X*	c.121G>T; p.(Glu41*)	LP: PVS1, PM2_supp
3	**Corneal** scarring/**neovascularization**, CAT, MY	**DA**, **dysmorphic facial features**, **CHD**, **hip dysplasia**, **feeding difficulties**, palate anomaly, vesicoureteral reflux, skin pigment anomaly, scoliosis, **gross motor delay but normal cognition**	*USP9X*	c.1603dupA p.(Ile535Asnfs*11)	LP: PVS1, PM2_supp
4	**ARA**	**DA**, **dysmorphic facial features**, **HL**, connective tissue anomaly, arachnodactyly	*JAG1*	NM_000214.3:c.59dupT p.(Leu21Alafs*52)	LP: PVS1, PM2_supp
5	**ARA**, MY	**DA**, **HL**, **CHD**, liver dysfunction, pituitary macroadenoma (prolactinoma)	*JAG1*	Gene deletion: 3.01 Mb deletion of 20:10256130-13269324 (hg19)	P: 2A
6	**ARA**, **CGL**, **corectopia**, HY	**DA**, **heart murmur**, PCOS	*JAG1*	NM_000214.3: c.2419G>T p.(Glu807*)	P: PVS1, PM2_supp, PP1
7	**ARA**, megalocornea	**Dysmorphic facial features**, cognitive impairment	*CDK13*	NM_003718.5:c.2252G>A p.(Arg751Gln)	LP: PS1, PM1, PM2_supp, PP3
8	**GL**, CAT, ONH, mild MI	**DA**, **HL**, **dysmorphic facial features**, **RU**, **CHD**, palate anomaly	*BCOR*	NM_001123385.2:c.3350_3360dup p.(Ala1121Glnfs*42)	LP: PVS1, PM2_supp
9	**CGL**, **IH**, **polycoria**, CAT, mild MI	**DA**, **HL**, rheumatoid arthritis, connective tissue anomaly	*HCCS* and *AMELX*	Gene deletions: 4.08 Mb deletion of X:7370404-11445756 (hg19)	P: 2A

ARA: Axenfeld–Reiger anomaly; ASD: anterior segment dysgenesis; CAT: cataract; CGL: congenital glaucoma; GL: glaucoma; HY: hyperopia; MI: microphthalmia; MY: myopia; DA: dental anomalies; HL: hearing loss; PCOS: polycystic ovarian syndrome; RU: redundant umbilicus. ACMG criteria [15,16]. **Features in bold are typical for ARS types 1 or 3.**

## Data Availability

There are no other data associated with this manuscript.

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
