# Peer review of "Alternative Genetic Diagnoses in Axenfeld–Rieger Syndrome Spectrum"

_genes, 2023, doi:10.3390/genes14101948_

Round 1

Reviewer 1 Report

This paper describes a series of patients with Axenfeld-Rieger Anomaly and identifies a genetic diagnoses in genes other than the two predominant causes of this collection of syndromes.  The authors describe the clinical features in each patient in detail. They demonstrate expansion of the phenotypes associated with each gene, connecting USPX9  to Axenfeld-Rieger anomaly for the first time and highlighting overlapping features with Alagille syndrome in families with JAG1 variants.

The paper is clearly written and easy to follow and adds substantial new knowledge to the field.

Minor suggestions:

1. The abbreviation ARS is not defined in the main text (only in the abstract)

2. I think there is a word missing from the end of the 3rd sentence in the Introduction, after the brackets describing umbilical features of ARS type 1. Without the brackets, the sentence currently reads  "......universally associated with dental and umbilical." 

3. First line of Methods: Should it be "Institutional Review Board" (instead of institutional research board)?

4. The methods would benefit from description of the variant filtering criteria applied. The bioinformatics pipeline is briefly referred to in another publication, but please outline the specific criteria applied for variant prioritisation in the current study. Were the ACMG guidelines applied to every variant identified in each individual, or were other filters (MAF, CADD etc) applied first?

5. Please define GL in Table 1. Is this glaucoma other than congenital glaucoma?

6. Paragraph 3 of the discussion describes Alagille syndrome, but doesn't connect it to the JAG1 gene until the last sentence. I recommend noting this earlier in the paragraph.

Author Response

Thank you for your comments. Please see below our responses to each comment in bold.

Comments and Suggestions for Authors (Reviewer 1):

This paper describes a series of patients with Axenfeld-Rieger Anomaly and identifies a genetic diagnoses in genes other than the two predominant causes of this collection of syndromes.  The authors describe the clinical features in each patient in detail. They demonstrate expansion of the phenotypes associated with each gene, connecting USPX9  to Axenfeld-Rieger anomaly for the first time and highlighting overlapping features with Alagille syndrome in families with JAG1 variants.

The paper is clearly written and easy to follow and adds substantial new knowledge to the field.

Author response: Thank you!

Minor suggestions:

  1. The abbreviation ARS is not defined in the main text (only in the abstract)

Author response: This has been added.

  1. I think there is a word missing from the end of the 3rd sentence in the Introduction, after the brackets describing umbilical features of ARS type 1. Without the brackets, the sentence currently reads "......universally associated with dental and umbilical."

Author response: We added the term ‘features’ to the end of this sentence.

  1. First line of Methods: Should it be "Institutional Review Board" (instead of institutional research board)?

Author response: Yes! This was corrected.

  1. The methods would benefit from description of the variant filtering criteria applied. The bioinformatics pipeline is briefly referred to in another publication, but please outline the specific criteria applied for variant prioritisation in the current study. Were the ACMG guidelines applied to every variant identified in each individual, or were other filters (MAF, CADD etc) applied first?

Author response: Variants were filtered for rare variants first, then variants within the gene groups indicated (ASD and OMIM) were reviewed. This was added to the manuscript.

  1. Please define GL in Table 1. Is this glaucoma other than congenital glaucoma?

Author response: Yes, GL is glaucoma. This was added to the legend.

  1. Paragraph 3 of the discussion describes Alagille syndrome, but doesn't connect it to the JAG1 gene until the last sentence. I recommend noting this earlier in the paragraph.

Author response: The genetic causes of Alagille syndrome were added to the first sentence of this paragraph.

Reviewer 2 Report

This work deals with alternative genetic diagnoses in patients with Axenfeld-Rieger Spectrum diagnosis, negative to FOXC1/PITX2 sequencing/CNV analysis. 

The work, mostly descriptive, reports alternative diagnosis in patients with this spectrum. It is useful to suggest alternative diagnosis to clinicians. 

Author Response

Comments and Suggestions for Authors (Reviewer 2):

This work deals with alternative genetic diagnoses in patients with Axenfeld-Rieger Spectrum diagnosis, negative to FOXC1/PITX2 sequencing/CNV analysis.

The work, mostly descriptive, reports alternative diagnosis in patients with this spectrum. It is useful to suggest alternative diagnosis to clinicians.

Author response: Thank you!

Reviewer 3 Report

This paper describes 9 patients with anterior chamber eye anomalies of Axenfeld-Rieger syndrome (ARS)-type plus associated other abnormalities, who have likely pathogenic alterations in genes other than the two most commonly causative ones (PITX2 and FOXC1).  It seems an excellent report, and will be valuable to anyone seeing such a person or researching this problem.

I have only 4 minor suggestions.

1.        Line 53-55  The origin of the 70% figure for proportion of ARS who have mutations in the 2 most common genes involved, needs referencing.

2.       Table 1  Please add a general reference for the ACMG criteria, so that anyone unfamiliar with this can look-up the abbreviations which the authors here use.   

3.       Line 90-96.  There is a formatting problem, in that it is unclear where the Table 1 legend finishes and the text restarts.  Please add an appropriate line space.

4.       Line 168  It may be premature to be so definite that the CDK13 or BCOR gene variants ‘explained a single family each’ , particularly if CDK13 variants have not previously been associated with anterior chamber anomalies.  It would seem better merely to state ‘….were identified in a single family each…’

Author Response

Comments and Suggestions for Authors (Reviewer 3):

This paper describes 9 patients with anterior chamber eye anomalies of Axenfeld-Rieger syndrome (ARS)-type plus associated other abnormalities, who have likely pathogenic alterations in genes other than the two most commonly causative ones (PITX2 and FOXC1).  It seems an excellent report, and will be valuable to anyone seeing such a person or researching this problem.

Author response: Thank you!

I have only 4 minor suggestions.

  1. Line 53-55 The origin of the 70% figure for proportion of ARS who have mutations in the 2 most common genes involved, needs referencing.

Author response: Reference was added.

  1. Table 1 Please add a general reference for the ACMG criteria, so that anyone unfamiliar with this can look-up the abbreviations which the authors here use.  

Author response: Reference cited in methods was added to Table 1 legend.

  1. Line 90-96. There is a formatting problem, in that it is unclear where the Table 1 legend finishes and the text restarts.  Please add an appropriate line space.

Author response: Layout was revised so Table 1 fills all of page 3

  1. Line 168 It may be premature to be so definite that the CDK13 or BCOR gene variants ‘explained a single family each’ , particularly if CDK13 variants have not previously been associated with anterior chamber anomalies.  It would seem better merely to state ‘….were identified in a single family each…’

Author response: This was changed.